# The Utility of Heartrate and Heartrate Variability Biofeedback for the Improvement of Interoception across Behavioural, Physiological and Neural Outcome Measures: A Systematic Review

**DOI:** 10.3390/brainsci14060579

**Published:** 2024-06-05

**Authors:** Lettie Wareing, Megan Rose Readman, Matthew R. Longo, Sally A. Linkenauger, Trevor J. Crawford

**Affiliations:** 1Department of Psychology, Fylde College, Lancaster University, Bailrigg, Lancashire LA1 4YF, UK; m.readman1@lancaster.ac.uk (M.R.R.); s.linkenauger@lancaster.ac.uk (S.A.L.); t.crawford@lancaster.ac.uk (T.J.C.); 2Department of Primary Care and Mental Health, The University of Liverpool, Waterhouse Building Block B, 2nd Floor, Liverpool L69 3GL, UK; 3National Institute of Health Research Applied Research Collaboration North-West Coast, The University of Liverpool, Waterhouse Building Block B, 2nd Floor, Liverpool L69 3GL, UK; 4School of Psychological Sciences, Birkbeck, University of London, Malet Steet, Torrington Square, Bloomsbury, London WC1E 7JL, UK; m.longo@bbk.ac.uk

**Keywords:** interoception, heartrate variability, biofeedback, heartrate variability biofeedback, interoceptive inference

## Abstract

Interoceptive dysfunctions are increasingly implicated in a number of physical and mental health conditions. Accordingly, there is a pertinent need for therapeutic interventions which target interoceptive deficits. Heartrate and heartrate variability biofeedback therapy (HR(V)-BF), interventions which train individuals to regulate their cardiovascular signals and constrain these within optimal parameters through breathing, could enhance the functioning of interoceptive pathways via stimulation of the vagus nerve. Consequently, this narrative systematic review sought to synthesise the current state of the literature with regard to the potential of HR(V)-BF as an interoceptive intervention across behavioural, physiological and neural outcome measures related to interoception. In total, 77 papers were included in this review, with the majority using physiological outcome measures. Overall, findings were mixed with respect to improvements in the outcome measures after HR(V)-BF. However, trends suggested that effects on measures related to interoception were stronger when resonance frequency breathing and an intense treatment protocol were employed. Based on these findings, we propose a three-stage model by which HR(V)-BF may improve interoception which draws upon principles of interoceptive inference and predictive coding. Furthermore, we provide specific directions for future research, which will serve to advance the current knowledge state.

## 1. Introduction

### 1.1. Interoception

Interoception, the sensing of internal signals arising from one’s own body [1] is vital for physical and mental health [2,3]. Specifically, homeostasis, the process of optimally regulating our internal milieu, is contingent upon one’s ability to accurately sense and integrate afferent signals from across the body within the brain [4]. Similarly, the ability to experience and regulate emotions depends upon our perceptions of our visceral sensations [5]. Unsurprisingly, therefore, interoceptive deficits have gained traction as a transdiagnostic process underlying a variety of clinical conditions ranging from chronic pain conditions [6,7], to anxiety and depression [8], to eating disorders [9].

Nevertheless, considerable debate surrounds both the definition and measurement of interoception (see [10] or [11] for a discussion). Some definitions regard interoception as a bi-directional process including both ascending afferent interoceptive pathways from body to brain implicated in the ‘sensing’, ‘interpreting’, and ‘integrating’ of internal bodily signals, but also descending efferent pathways from the brain to the body implicated in ‘regulating’ internal states [12]. In contrast, other definitions recognise the role of bi-directional communication in interoception, viewing sensation as arising from the integration of afferent sensory information with top-down expectations about these signals, but regard regulation via efferent pathways as distinct from this [10].

The lack of a clear definition of interoception has led to challenges in defining valid and reliable measures of interoception. Nevertheless, one point of agreement amongst several modern definitions of interoception is that interoception involves conscious and non-conscious processes [10,12,13]. Consequently, measures of interoception can span different levels of processing from the conscious awareness of visceral signals to the non-conscious neural representation of interoception [14].

### 1.2. Measuring Interoception

Research studies predominantly measure interoception within the cardiovascular system [11]. Cardiovascular interoception begins with the relaying of afferent signals from the vagus nerve to the nucleus of the solitary tract where signals are subsequently passed on to higher-level brain regions, including the insular and anterior cingulate cortices [15]—the ‘interoceptive hubs’ of the brain [16,17]. Accordingly, neural measures of cardiovascular interoception include correlations between brain activation, or structure, and performance on behavioural interoceptive tasks (e.g., [18]); patterns of brain activation when attending to interoceptive signals (e.g., [19]); but also, heartbeat-evoked potentials (HEP; e.g., [20]), electrophysiological markers of heartbeat detection or attention [21].

Conscious awareness of interoceptive sensations is typically measured using behavioural tasks or questionnaires. At this level of processing, further dimensions have been proposed [1] (though see [13]) with behavioural tasks typically classified according to the interoceptive dimension they are thought to map onto. Specifically, Garfinkel, et al. [1] proposed three dimensions of interoception: interoceptive accuracy assesses the reliability of our interoceptive perceptions using objective behavioural measures, interoceptive sensibility measures our degree of self-confidence in our interoceptive perceptions, and interoceptive awareness provides a measure of the corroboration between interoceptive accuracy and awareness (i.e., metacognitive awareness of one’s interoceptive abilities). A classic behavioural measure is the heartbeat counting task [22], which requires individuals to count how many heartbeats they can feel over a specific time duration. This task provides a measure of interoceptive accuracy (by comparing participants’ estimates to their veridical heartbeat over the same time period), but also awareness and sensibility, if a measure of participants’ confidence in their estimates is obtained. Alternative behavioural measures also exist including both task-based (e.g., the heartbeat discrimination task [23]) and questionnaire-based methods or paradigms (e.g., the interoceptive accuracy scale [24]).

Yet, whilst debate surrounds whether regulation via efferent pathways can be incorporated within the definition of interoception [12], or distinct from this [10], interoception is nonetheless implicated in regulatory processes. Specifically, to engage in optimal regulation of the body’s physiological (and emotional) state, interoceptive afferents must be accurately perceived and integrated within the brain [20,25,26,27].

Top-down regulation of the body is thought to be coordinated by the central autonomic network [28]. This is a cluster of brain regions including not only the insular and anterior cingulate cortices, but also regions such as the prefrontal cortex [29] whose function is to pre-empt and respond to changes in the body’s physiological state in order to maintain homeostasis [30]. Within the cardiovascular system, the central autonomic network regulates cardiac activity by sending efferent signals via the nucleus of the solitary tract to thoracic (sympathetic) or vagus (parasympathetic) nerves within the sinoatrial node of the heart [28], which produce increases or decreases in the heartrate, respectively [31]. A simplified depiction of the efferent and afferent pathways involved in the detection and regulation of cardiovascular activity by the central autonomic network can be found in Figure 1.

Accordingly, one way in which interoception might be indexed indirectly could be to measure variables associated with optimal autonomic functioning, such as physiological measures of central autonomic activity [33]. With respect to this, cardiovascular autonomic activity is typically indexed via heartrate variability (HRV), the beat-to-beat variation in the R-R interval (the period between heartbeats) caused by fluctuations in the activation of the parasympathetic or sympathetic nervous systems [34]. Specifically, different time and frequency domain parameters of HRV can provide insights into the degree of parasympathetic and sympathetic influences over the heartrate, as well as overall autonomic activity [35]. For example, high-frequency HRV as well as the time-domain measure root mean square of successive differences (RMSSD) reflect parasympathetic activity and hence regulation of the heart by the central autonomic network [36]. In contrast, total power and the standard deviation of N-N intervals (SDNN) are measures of the total variance in the heartrate signal [37] and thus represent overall autonomic activity. At rest, higher levels of HF-HRV and RMSSD as well as SDNN and total power are desirable with lower levels of these indices typically observed in clinical populations [38].

Whilst these measures do not directly index interoception, if an individual’s autonomic activity, particularly their levels of parasympathetic or sympathetic activation, is optimal in a given context, then it could be that they are able to appropriately able to detect, integrate, and respond to interoceptive afferents. Supporting this assumption, higher resting RMSSD positively correlates with interoceptive accuracy [39] and high-frequency HRV positively correlates with measures of interoceptive sensibility [40]. Similarly, lower SDNN has been observed in medicated and medication-free individuals with conditions in which interoceptive deficits have been implicated, including depression [41], anxiety [42], and post-traumatic stress disorder [43]. Hence, these findings provide support for a relationship between higher interoceptive abilities and better autonomic regulation.

However, currently, the insights into interoception that can be provided by studying indices of autonomic activity are largely overlooked. This is possibly because autonomic indices such as HRV can be influenced by a variety of extraneous factors, including sleep [44], posture [45], respiration rate [46], body weight [47], and caffeine intake [48], amongst other factors. Therefore, the reliability of physiological indices as interoceptive measures may be limited. Nevertheless, standardised guidelines have been published that outline how best to control for extraneous influences on the HRV signal [49]. After controlling for these extraneous factors, if an individual’s physiological indices improve from baseline to post-intervention, then this may indicate that central autonomic regulation and, therefore, the sensing and integration of interoceptive afferents may have possibly been enhanced by the intervention.

### 1.3. Heartrate (Variability) Biofeedback

Whilst some interoceptive therapies are in development, such as interoceptive exposure therapy [50], or mindfulness [51], there is mixed evidence to support their efficacy in improving both interoception and mental health symptoms across different conditions [52]. Therefore, there is a need to consider other interventions which have the potential to restore optimal interoception. One such candidate interoceptive intervention is biofeedback therapy, which involves training individuals to use biofeedback (the online measurement and displaying of physiological processes [53]) to constrain physiological signals within optimal parameters [54]. Specifically, the use of heartrate variability or heartrate biofeedback therapies (collectively, HR(V)-BF) has become increasingly popular. In these therapies, individuals may be shown their actual HRV or HR signal trace on a display (e.g., [55]), or shown a scene which changes in line with an individual’s HR or HRV signal (e.g., [56]). In either of these presentation modes, individuals are taught techniques to control the signal or scene and thus maintain HR or HRV within desired parameters.

The goal of HR(V)-BF therapy is to enhance HRV at rest and induce optimal autonomic regulation. HR(V)-BF therapy was developed based on observations that breathing at a frequency of around 0.1 Hz (~six breaths per minute; [57]) maximises the amplitude of heartrate oscillations [58]. More specifically, the heartrate rises with inspiration and decreases with expiration, a phenomenon known as respiratory sinus arrhythmia (RSA; [59]). Typically, these changes in heartrate and respiration are not entirely aligned [60]; however, by breathing at a rate of 0.1 Hz, it is thought that heartrate and respiratory oscillations are brought in phase [61]. In turn, as the heartrate rises with inspiration, blood pressure also rises, and vice versa [62]. These changes in blood pressure trigger activation of the baroreflex which responds to blood pressure increases or decreases by activating the parasympathetic or sympathetic nervous systems to decrease, or increase the heartrate, respectively [63]. Thus, by breathing at 0.1 Hz, a resonance effect is generated in which both RSA and baroreflex sensitivity are enhanced [64].

Importantly, RSA is thought to be regulated by parasympathetic efferents which are transmitted via the vagus nerve [65] (though see [66]), a cluster of efferent parasympathetic nerve fibres, but also afferent nerve fibres responsible for communicating visceral afferents to the brain [67]. Accordingly, by breathing at one’s resonance frequency, HR(V)-BF is thought to stimulate the vagus nerve, increasing both the intensity of afferent signalling to the brain, but also enhancing efferent parasympathetic activity through triggering increases in RSA [60]. Consequently, through stimulating the vagus nerve and thus increasing afferent signalling to the brain, it could be hypothesised that HR(V)-BF may improve interoception by enhancing the detection and integration of interoceptive afferents.

Support for the relationship between vagus nerve stimulation and interoception comes from studies of both implanted and transauricular vagus nerve stimulation interventions. Vagus nerve stimulation has been shown to increase the functional connectivity of interoceptive brain regions including the cingulate cortex and anterior insula [68,69]. Moreover, vagus nerve stimulation can enhance the heartbeat-evoked potential, an effect which was localised to the insula, as well as regions of the central autonomic network and somatosensory cortex [70]. Hence, stimulating the vagus nerve has direct effects on neural indicators of, and the functional connectivity of brain regions implicated in, interoception. Critically, concurrent increases in both behavioural measures of interoceptive accuracy and the heartbeat-evoked potential have been found to occur post-vagus nerve stimulation [71,72]. Consequently, stimulating the vagus nerve appears to impact interoception at both conscious and non-conscious levels. Similarly, enhancements of the heartbeat-evoked potential have also been observed during HR(V)-BF [73] and hence, given its similar proposed mechanisms of action, it is highly possible that HR(V)-BF may lead to improvements in interoception. Importantly, HR(V)-BF is non-invasive, relatively cheap, and easy to administer making it a viable alternative to vagus nerve stimulation [74].

### 1.4. The Current Review

To date, there has been no attempt to summarise research regarding the effects of HR(V)-BF therapy on outcome measures related to interoception. Consequently, the purpose of this systematic review is to determine the current state of knowledge concerning the effects of HR(V)-BF therapy on interoception and to provide directions for future research that may help to advance the field, based on these findings. To do this, we will consider the effects of HR(V)-BF on interoceptive outcome measures from behavioural self-report and neural measures, but also select physiological outcomes. Specifically, physiological outcomes which reflect vagal (parasympathetic) mediation of cardiovascular activity (e.g., RMSSD), as well as those which reflect general enhancements in autonomic activity or afferent signalling (e.g., baroreflex sensitivity, or total power) will be reviewed.

## 2. Methodology

### 2.1. Transparency and Openness

This review was conducted in accordance with the Preferred Reporting Items for Systematic Reviews and Meta-Analyses (PRISMA; [75]) guidelines and was pre-registered on PROSPERO (ID CRD42022370067). All supporting information including the anonymised protocol, search strategy, screening criteria, screening decision logs, quality assessment, and coding of methodological characteristics can be found in the Appendix A on the Open Science framework available at https://osf.io/9jsy5/, accessed on 30 May 2024.

### 2.2. Eligibility Criteria

Studies were eligible if they (a) constituted an original experimental investigation of HR(V)-BF therapy, (b) were written in the English language, (c) included a control/comparator group, and (d) included at least one interoceptive outcome measure (behavioural, physiological, or neural) taken at rest, pre- and post-intervention. Studies were not excluded if the HR(V)-BF therapy was used in conjunction with another form of biofeedback or psychotherapy; however, this was noted and considered when evaluating findings.

We defined HR(V)-BF therapy as the use of HR or HRV biofeedback for clinical purposes. Specifically, studies which involved using HR(V)-BF to train individuals in techniques, which allowed them to optimally regulate their HR or HRV, in order to improve HRV or clinical symptoms were included. By this definition, studies utilising healthy samples could be included provided HR(V)-BF therapy was used with the purpose of enhancing physiological functioning or wellbeing. However, studies where HR or HRV biofeedback was not used therapeutically to train participants to constrain physiological signals within optimal parameters were excluded. For example, studies where biofeedback was used to train participants to produce non-specific increases or decreases in the heartrate with no clinical purpose, or those studies where biofeedback was used to teach participants to sense how many heartbeats have occurred over a specific time duration without requiring them to manipulate the heartrate signal were not included.

In addition, we excluded studies where the outcome was measured whilst individuals engaged in paced breathing or were under stress (e.g., [76]), as we were interested in resting-state changes in outcomes. Studies which measured an agreed physiological outcome, but only reported this measure using normalised HRV units (i.e., dividing a frequency measure by the sum of spectral power; [77]) were also excluded (e.g., [78]) given that the interpretation of normalised units is highly debated [35,77]. Where studies included both normalised and non-normalised outcome measures, only the non-normalised outcomes are discussed. Studies which failed to report *p*-values for both intervention and comparator groups were excluded as this precluded reliable conclusions as to the effects of the intervention.

Clarification as to what constituted behavioural, physiological, or neural measures of interoception can be found in Table 1. Physiological measures were chosen on the basis that they reflected enhanced signalling, communication, or regulation of interoceptive afferents and thus could provide an indirect measure of interoception. Justifications for the decision to include, or exclude, different measures of cardiovascular activity can be found in Appendix A. No restrictions were placed on the year of dissemination or sample population. The grey literature returned from database searches was included providing all criteria were met.

### 2.3. Information Sources

A comprehensive literature search using Academic Search Ultimate, PsycINFO, MEDLINE Complete, CINAHL, Scopus and Web of Science was conducted on 28 October 2022. Databases were selected to (1) reflect the multidisciplinary nature of the research question and (2) ensure the grey literature was captured. To ensure the comprehensiveness of the search, four search strings were compiled. The first included search strings relating to biofeedback, and the remaining three strings corresponded to a different interoceptive measurement modality (behavioural, physiological, or neural). The search strings applied to each database differed only in the database-specific definition terms included. It was noted that the pre-registered search strings did not include anterior cingulate cortex search terms; therefore, searches were re-conducted on 28 November 2022 with the addition of three search terms (“anterior cingulate cortex”, “ACC”, and “cingulate cort*”). The full search strategy for each database can be found in Appendix A. To ensure all relevant records were captured, prior to submitting, searches were re-conducted on 14 November 2023. The same search strings and databases were used for the updated search, but the publication date was limited to between November 2022 and November 2023.

Backwards citation searching of key papers identified in the pre-registered protocol occurred on 6th December 2022. Citations were downloaded from Scopus, except for [80], in which citations were not available via this database and as such were manually downloaded using Google Scholar. It should be noted that some key papers used for backwards citation searching did not meet all inclusion criteria and were not included in the review.

### 2.4. Selection Process

The screening process is depicted in Figure 2. Criteria used for both title and abstract and full-text screening can be found in Appendix A. CADIMA ([81] https://www.cadima.info/, accessed October 2022) was used for de-duplication and title-and-abstract screening of database and backwards search results. However, due to software issues, full-text screening was conducted using Microsoft Excel 2019 for all papers. Two reviewers screened all papers at the title-and-abstract level simultaneously and were blinded to the others’ decisions until screening was complete. Inter-rater agreement at the title-and-abstract level was substantial (93.80%, Cohen’s k = 0.66). Given this, at full-text screening, the first 10 percent of papers were screened independently by the two reviewers. After ensuring consistency was still high (89.58%, Cohen’s k = 0.64), the remaining papers were split between the reviewers and screened independently. Any inconsistencies at both the title-and-abstract and full-text levels were resolved via discussion between reviewers. Due to the high level of consistency on these ratings, and when performing the initial screening and backward screening stages, for the re-conducted searches, it was decided that only a random sample of 25% of papers (*n* = 13) at the full-text level would be screened by both reviewers, after which screening decisions were made by a single reviewer.

Of the 4536 papers screened (4177 initial search; 359 re-run search), 4470 (4122 initial search, 348 re-run search; 98.54%) failed to meet the inclusion criteria. The first reason for exclusion for all papers reviewed at the title-and-abstract and full-text levels are reported in Appendix A.

### 2.5. Quality Assessment

Quality assessment was conducted by two reviewers using the adapted version of the Downs and Black [82] checklist pre-registered in the protocol. The reviewers conducted quality assessments simultaneously and were blinded to the others’ decisions. Any disagreements were resolved by discussion between reviewers.

It was noted during the quality assessment that one of the included questions (Question 8) relating to the reporting of adverse events was not relevant to the types of study included in this review; therefore, this question was excluded from quality score calculations. The adapted quality assessment was scored out of 23 and ratings were categorised according to previously published criteria [83], adapted to the maximum score attainable given the applied checklist. Scores were thus ranked as Excellent (21–23), Good (15–20), Fair (14–10), or Poor (≤9).

The conference abstracts included were not assessed for quality, as their short word limit meant insufficient data for quality assessment could be obtained from the text. The maximum quality score achieved by an included paper was 20 (Range 11–20), with the average score being ‘Fair’, (M = 14.65, SD = 2.13). A breakdown of the criteria applied and scores for each item for all included papers can be found in Appendix A.

### 2.6. Data Extraction and Data Synthesis

Data extraction was conducted by one reviewer using Microsoft Excel and checked by another reviewer. Data extracted included the following: number of participants, mean age of participant groups, interoceptive measurement modality and specific interoceptive outcome measure(s) used, timepoints of measurement, length of HRV measure, means and standard deviations, and directions of main effects. In addition, we reported methodological characteristics including the type of intervention (HR- or HRV-BF), method of delivery, biofeedback device used, session frequency, and breathing protocol used.

The method of delivery was classified as “standard HR(V)-BF” if it involved following a practitioner-guided HR(V)-BF protocol at a specific venue. Other studies used HR(V)-BF in conjunction with other therapies or biofeedback, non-traditional HR(V)-BF formats (e.g., virtual reality), or solely home practice protocols and were recorded as such. Characteristics of the comparator were also reported. Studies which did not report significance values were not included in the final review. Additionally, we categorised breathing protocols using an adapted version of previously published criteria [84]. Specifically, breathing protocols were classed as “optimal” if the individual’s exact resonance frequency was determined at the start of the intervention, in line with Lehrer, et al.’s [85] protocol; “individual” if resonance frequency was achieved by synchronising the breathing and heartrate waveforms in each session; “paced/preset” if participants were instructed to breathe at a specific breathing rate (e.g., six breaths per minute); and “unclear” if the breathing protocol used could not be determined or the study referenced a protocol but did not describe their own specific implementation of this protocol. Categorisations for each included study can be found in Appendix A.

Given the novelty of this review topic, we chose to conduct a systematic review without meta-analysis as this afforded space to explore the nuances of the findings and identify gaps in knowledge. Although meta-analyses can accommodate a degree of study heterogeneity, we felt that the significant variability amongst included studies with regard to the sample population, intervention duration, comparator group(s), intervention protocol (including whether resonance frequency breathing was used), and measurement of outcomes (e.g., whether short- or long-term HRV measures were used) would likely significantly decrease power [86] and consequently prevent meaningful group level comparisons from being drawn. Consequently, a qualitative systematic review of the data was conducted following published guidance [87].

## 3. Results

Seventy-seven studies met the inclusion criteria (see Appendix A for full extracted data). Forty-five studies utilised clinical populations (58.44%), whilst thirty-two used healthy populations (41.56%). Comparator groups included treatment-as-usual (n = 17, 20.00%), no-intervention (n = 35, 41.18%), active controls (n = 24, 28.24%), and sham/placebo biofeedback (n = 9, 10.59%) with some incorporating multiple comparators. Most studies used an optimal breathing protocol (n = 32, 41.56%), whilst others used individual (n = 15, 19.48%) or paced/preset (n = 21, 27.27%) protocols. Nine studies (11.69%) did not clearly report a breathing protocol. Intervention intensities varied from intense (i.e., more than one session per week; n = 28, 36.36%), to moderate (i.e., five or more weekly sessions; n = 25, 32.47%), to mild (i.e., less than five weekly sessions; n = 24, 31.17%). Intervention intensity was judged in relation to five sessions as this is the number of sessions originally specified in Lehrer, et al.’s [84] protocol. Given that home practice was not always reported in depth, particularly with respect to compliance, the intensity was judged only upon in-person sessions, except for solely home-intervention protocols.

For each outcome measure reported, a summary of the key methodological characteristics (based on the categorisation criteria used for data extraction and presented in the methodological characteristics table (Appendix A)) of studies observing each effect direction (effect of intervention, no effect, or another effect) can be found at the end of the section. In addition, for an overall summary of the results for each outcome, the reader is referred to Table 13 at the end of the results section.

### 3.1. Behavioural Measures

One study incorporated a behavioural outcome measure [73]. This found no effect of HRV-BF (using individualised resonance frequency breathing) or the electromyography feedback comparator on heartbeat counting task performance in healthy individuals. This study had a Good quality assessment score (19). Further characteristics of this study can be found in Table 2.

#### Conclusions—Behavioural Measures

HRV-BF may not improve interoception using behavioural outcomes. However, with only one study, conclusions are limited.

### 3.2. Physiological Measures

#### 3.2.1. High Frequency HRV (HF-HRV)

HF-HRV (0.15–0.40 Hz) reflects mostly parasympathetic activity and respiratory rates of 9–24 breaths per minute [35]. At rest, it is primarily composed of respiratory sinus arrhythmia [36].

HF-HRV was measured in 46 studies. The methodological characteristics of these studies can be found in Table 3. Thirty-three studies reported no effect of HR(V)-BF or the comparator at post-intervention. Of these, 23 [88,89,90,91,92,93,94,95,96,97,98,99,100,101,102,103,104,105,106,107,108,109,110] recruited clinical populations and 10 ([111,112,113,114], ([115], study 1, study 2), ([116], institutionalised group), [117,118,119]) studied healthy populations.

Eight studies (four using clinical populations) found HF-HRV pre- to post-intervention increased [56,120], ([116], non-institutionalised group), [121,122,123,124,125] and at 3-month follow-up [56] for HRV-BF, but not comparator groups. However, whilst Kudo, et al. [124] and de Souza, et al. [116] (non-institutionalised group) found HF-HRV increased pre- to post-intervention more in the HRV-BF group, there was no difference in post-intervention HF-HRV between HRV-BF and comparator groups.

Some findings were more mixed. Lin, et al. [126] found higher HF-HRV post-intervention in the comparator than HRV-BF group. In contrast, Caldwell and Steffen [127] and Nashiro, et al. [128] found HF-HRV decreased over time for both HR(V)-BF and the comparator and Dziembowska, et al. [129] found HF-HRV decreased between pre- and post-intervention only for the HRV-BF group, whereas Yu, et al. [130] found lower HF-HRV in the HRV-BF group across timepoints relative to the comparator group.

Taken together, the majority of studies have found that HRV-BF has little effect on HF-HRV, especially amongst clinical populations. There was very little difference between studies finding an effect and those which did not in quality scores, indicating that study quality is not related to the likelihood of observing an effect.

#### 3.2.2. Low-Frequency HRV (LF-HRV)

LF-HRV (0.04–0.15 Hz; [35]) reflects both parasympathetic and sympathetic influences [131] as well as baroreceptor activity [132]. At slower breathing rates, parasympathetic activity can shift to the LF-HRV band at rest [64]. The methodological characteristics of all studies utilising this outcome measure can be found in Table 4.

Forty-one studies measured LF-HRV. Nineteen (ten using clinical populations) found higher post-intervention LF-HRV in the HRV-BF group than comparators or relative to pre-intervention [56,93,94,95,99,100,106,110,112,113,115], ([119], study 1), [121,123,126,128,129,130,133]. Five of these incorporated a follow-up, with one study finding no significant difference between HRV-BF and comparator groups from pre-intervention to a 4-week follow-up [119] whereas others found effects remained at 1 month [56,93,94], 3 months [56] and at 1 year [130]. Interestingly, Ratajczak, et al. [121] found that a significant interaction between group and time-point was only observed after removing those participants from the HRV-BF who failed to achieve resonance (i.e., in the per-protocol sample), suggesting resonance breathing may be important for this effect.

Three studies (all clinical populations) observed only a time effect [107,122,124]. However, for Tastchl, et al. [107], post hoc tests found only significant increases over time for HRV-BF. In two studies, the comparator engaged in slowed breathing, suggesting slowed respiration could underlie the time effect. Supporting this idea, Lin [113] found LF-HRV increased for HRV-BF and relaxation training groups (who engaged in slow breathing) but not no-intervention controls.

In contrast, 18 studies (13 with clinical populations) observed no effect on LF-HRV ([88,89,90,91,92,96,97,98,101,104,105,109,111], ([115], study 2), [117,118,125,127]).

Finally, Whited, et al. [114] observed higher LF-HRV in both pre- and post-intervention for the HRV-BF group relative to comparators.

Overall, results with regard to the effects of HR(V)-BF on LF-HRV are split. Interestingly, studies which found no effect had a higher proportion of clinical populations, suggesting this may influence the likelihood of finding an effect. In addition, some of the findings discussed suggest whether resonance breathing was used may be influential. Studies which found an effect had a slightly higher quality score, possibly indicating an effect of study quality on the outcome observed.

#### 3.2.3. Root Mean Square of Successive Differences (RMSSD)

RMSSD is a measure of the variation in heartrate that occurs between heartbeats [36]. This time-domain measure is calculated by first squaring the time difference between individual heartbeats. These values are then averaged and RMSSD is the square root of this average [134]. RMSSD primarily measures parasympathetic activity and is therefore highly correlated with HF-HRV [135]. A summary of the methodological characteristics of all studies using this outcome variable can be found in Table 5.

Forty-two studies measured RMSSD. Twenty-eight (twelve using clinical populations) observed no significant changes for intervention or comparator groups ([55,88,90,97,98,102,103,105,107,111,113,114], ([115], study 1), [117,118,119,126,128,130,136,137,138], ([139], study 1, study 2, study 3), [140,141,142]).

Thirteen studies (seven with clinical populations) found that RMSSD was higher for the HRV-BF group compared to pre-intervention, or the comparator, at post-intervention ([56,106,109], ([116], non-institutionalised group), [121,122,125,143,144,145,146,147,148]). However, for Prabhu, et al. [147], RMSSD was only greater at post-intervention in the HRV-BF groups than in the comparator when the intervention was given pre-knee arthroplasty operation, and not when the intervention was delivered to the same participants post-operation. In addition, Tinello, et al. [148] only observed significant increases in RMSSD from pre- to post-intervention following HRV-BF (but not an active control) in individuals with low baseline RMSSD, whereas those participants with high baseline RMSSD showed no change from pre- to post-intervention in either group. However, these analyses were only exploratory. Interestingly, Chang, et al. [56] found no difference between RMSSD in HRV-BF and comparator groups at a 1-month follow-up, but significantly greater RMSSD in the HRV-BF group at a follow-up of 3 months.

In contrast, de Souza, et al. [116] (institutionalised group) observed a significant time effect with RMSSD increasing significantly in both the HRV-BF group and the comparator.

Overall, the most consistent finding is that HRV-BF has little effect on RMSSD; however, findings suggest that the baseline level of RMSSD or baseline conditions may be influential. Quality scores were similar for both outcome directions, suggesting that quality does not affect the outcome observed. brainsci-14-00579-t005_Table 5Table 5Summary of the key features of included studies utilising RMSSD, including methodological characteristics (whether a clinical population was employed, type of comparator group, intensity of intervention, and type of breathing protocol) as well as mean risk of bias scores, grouped according to the effect observed. Methodological characteristics of studies finding each effect are reported as a proportion of all studies finding the same effect.Features of Included StudiesEffect of Intervention (*n* = 13)No Effect of Intervention (*n* = 28)Other Effect (*n* = 1) (i.e., Time or Effect of Comparator)Clinical population (%)53.8542.860.00Comparator Group (%) ^a^NI—23.57TAU—15.38 AC—42.86S/P—14.29NI—36.37TAU—18.19 AC—30.30S/P—15.16NI—0.00TAU—0.00AC—100.00S/P—0.00Intervention Intensity (%)Mild—15.39Moderate—23.08Intense—61.54Mild—57.14Moderate—25.00Intense—17.86Mild—0.00Moderate—0.00 Intense—100.00Resonance Frequency Breathing (%)Overall RF—53.85Optimal—30.77Individual—23.08Paced/preset—23.08Unclear—23.08Overall RF—57.14Optimal—46.43Individual—10.71Paced/preset—35.71Unclear—7.14Overall RF– 100.00Optimal—0.00Individual—100.00Paced/preset—0.00Unclear—0.00Risk of Bias14.45 ^b^14.3912.00Note. NI = No intervention, TAU = intervention-as-usual, AC = active control, S/P = sham/placebo. ^a^ Some studies used more than one comparator group; therefore, proportion has been calculated out of the total number of comparators for the main effect. ^b^ Risk of bias scores were not available for Vagedes, et al. [144] or Siepmann, et al. [146] as these were conference abstracts; therefore, the mean is calculated based on the remaining seven studies.

#### 3.2.4. Standard Deviation of N-N Intervals (SDNN)

SDNN reflects overall autonomic nervous system activity and hence is highly correlated with total power [37]. A summary of the methodological characteristics of all the studies using this outcome measure can be found in Table 6.

SDNN was included in 51 studies. Twenty-six studies (seventeen with clinical populations) observed higher SDNN at post-intervention/follow-up in the HRV-BF group relative to baseline, or comparators ([56,91,93,94,95,99,105,106,109,112,113], ([115], study 1), ([116], non-institutionalised group), [121,122,124,125,126,127,130], ([139], study 2), [140,142,143,149,150]). These effects held after controlling for confounds including age, session compliance, blood pressure, body mass index, and medications [91,124,130,142], and non-compliance [91]. However, despite finding a significant increase pre- to post-intervention in the HRV-BF group only, Krempel and Martin [105] did not find a difference between HRV-BF and an autogenic training comparator at post-intervention.

With regard to those finding an effect on SDNN, some interesting observations were made. Yu, et al. [130] observed an effect only after controlling for breathing rate and medication and Limmer, et al. [91] found effects were only present in short-term (5–10 min), but not long-term (24-h) recordings. Notably, Ratajczak, et al. [121] found that whilst pre- to post-intervention increases in SDNN were present in both groups when considering the full sample, when excluding participants who did not achieve resonance frequency from the HRV-BF group, training effects were only significant in the HRV-BF group. Similarly, Limmer, et al. [91] found a significant interaction in the HRV-BF group after controlling for compliance and other confounds. Furthermore, Chang, et al. [56] observed no significant differences in SDNN between groups at 1-month follow-up, but significantly higher SDNN in the HRV-BF group at a follow-up of 3 months. Intriguingly, Tinello, et al. [148] found that SDNN increased significantly from pre- to post-intervention in the HRV-BF, but not the comparator, only in those participants who had low baseline RMSSD. For those participants with high baseline RMSSD, no difference between pre- and post-intervention was observed for HRV-BF or comparator groups. However, these analyses were only exploratory.

In contrast, 21 studies (12 with clinical populations) observed no difference between pre- and post-intervention, or between HRV-BF and comparators at post-intervention ([55,88,89,90,92,96,97,98,100,103,107,111], ([115], study 2), ([116], institutionalized group), [117,118,119,141,145,151,152]). Notably, Vagedes, et al. [141] and Climov, et al. [151] used long-term instead of short-term measures.

Brinkmann, et al. [138] found a time effect with SDNN increasing over time for both groups, whereas Barnable [139] (study 1) found a significant increase in SDNN pre- to post-intervention in the comparator, but not the intervention group. After pooling the data from studies 1 and 2, Barnable [139] (study 3) found significant increases in both groups.

Finally, Whited, et al. [114] observed a group effect: SDNN was higher in the HRV-BF group across timepoints.

Overall, results for SDNN are mixed, though there is an indication that studies which controlled for compliance and covariance were more likely to observe effects. Consequently, HR(V)-BF may be effective but only under highly controlled circumstances. Quality scores do not appear to differ between effect directions suggesting this does not influence the outcome obtained.

#### 3.2.5. Percentage of Successive N-N Interval Pairs Differing by >50 ms (pNN50)

pNN50 is the proportion of successive R-R intervals (the time between two R waves on an electrocardiogram) which differ by more than 50 ms [36]. This measure is influenced by parasympathetic activity and correlates with both RMSSD and HF-HRV [35]. A summary of the methodological characteristics of all studies using this outcome measure can be found in Table 7.

Twelve studies included pNN50. Eight (five with clinical populations) found no effect of either HR(V)-BF or the comparator ([88,90,97,100,114], ([116], institutionalised group), [125,152]). In contrast, three studies (one including a clinical population) observed an effect. Specifically, de Souza, et al. [116] (non-institutionalised group) found higher pNN50 post-intervention after HRV-BF than the comparator; Bian, et al. [109] found pNN50 increased from pre- to post-intervention in the HRV-BF group, but not the comparator; whereas Kerr, et al. [119] observed greater increases in pNN50 from pre-intervention to a 4-week follow-up in the HRV-BF relative to the comparator, but not from pre- to post-intervention. In contrast, Lin, et al. [126] found higher pNN50 at post-intervention in the comparator.

Consequently, HRV-BF appears to have no effect on pNN50. Those who found an effect tended to be with healthy populations, perhaps suggesting an effect of population on the outcome, though this sample was small. Quality scores were similar for those that found an effect and those that did not.

#### 3.2.6. Total Power (TP)

TP is the sum of all energy across HRV frequency bands and hence higher TP at rest and therefore serves as an indicator of overall autonomic activity [37]. A summary of the methodological characteristics of studies employing this measure can be found in Table 8.

Overall, fifteen studies measured TP. Seven studies reported no effect of HRV-BF or comparator groups in healthy [98,111,118,129,145] and clinical [101,130] populations.

Eight studies (six with clinical populations) reported higher TP following HR(V)-BF than the comparator [56,93,106,109,110,121,122,123] with effects present at follow-up of 3 months [56].

The findings regarding TP are mixed with a similar number of studies finding an effect as those finding no effect. Interestingly, studies which found an effect had a higher quality rating, perhaps suggesting higher quality studies may be more likely to observe an effect with this outcome.

#### 3.2.7. Coherence Measures

Coherence measures the degree of resonance in the cardiovascular system or the extent to which the heartrate has become more ordered and localised around the resonance frequency [153]. The coherence ratio is a metric which accounts for variability in HRV over time and is thus calculated as peak power/(total power − peak power). In this equation, peak power is the power in the integral window and total power is the total power of the HRV signal [154]. A summary of the methodological characteristics of all studies using this outcome measure can be found in Table 9.

Five studies measured coherence or coherence ratios. Three (all clinical populations) observed that coherence (ratios) were higher at post-intervention for HR(V)-BF compared to comparators [129,155,156]. However, Amjadian, et al. [156] found that improvements in the HRV-BF group were only significantly greater than no-intervention controls, whereas coherence at post-intervention was comparable to a religious-based therapy comparator. In contrast, Rockstroh, et al. [55] observed no significant differences in changes in coherence ratios pre- to post-intervention between the HRV-BF groups and the comparator. Additionally, whilst Berry, et al. [157] reported a significant increase from pre- to post-intervention in the HRV-BF in coherence, the *p*-value was not below the significance threshold.

Consequently, there is a trend for HRV-BF to improve coherence measures, even within clinical groups. However, given the small number of studies using this outcome measure, further evidence is required. Interestingly, those studies which did not find an effect had a higher quality score, though this should be interpreted with caution given the small sample.

#### 3.2.8. Baroreflex Sensitivity (BRS)

BRS (sometimes referred to as baroreflex gain) provides a measure of baroreflex functioning in terms of the responsiveness of the baroreflex to changes in blood pressure [158]. As an index, BRS expresses how much the interbeat interval (the time interval between successive heartbeats) changes with every one unit change in blood pressure and is thus calculated as the increase in the interbeat interval in ms divided by the change in blood pressure over the same time period in mmHg [159].

Three studies (one clinical) measuring BRS [64,122,145] found that BRS increased post-intervention in the HRV-BF group relative to comparators, even when controlling for respiration [64], and when compared against slow abdominal breathing [122]. Whereas one study [108] found BRS did not change from pre- to post-intervention in the HRV-BF group or the comparator using participants with hypertension. In addition, improvements in BRS were close to, but did not reach the threshold for significance in the study of Lehrer and Vaschillo [160].

Overall, these findings indicate that HRV-BF could be used to improve BRS. However, this conclusion is limited by the small number of studies utilising this variable. In addition, studies which found an effect tended to have lower quality scores and so further caution should be taken when interpreting this outcome. A summary of the methodological characteristics of all studies using this outcome measure can be found in Table 10.

#### 3.2.9. Respiratory Sinus Arrhythmia (RSA)

RSA, the elevation of the heartrate that occurs with inspiration and the lowering of the heartrate occurring with expiration [161], is a centrally mediated phenomenon [162] and is reliant upon parasympathetic activity [163]. There are several ways of calculating RSA. These include the peak-valley approach which involves subtracting the shortest interbeat interval occurring during inspiration away from the maximum interbeat interval occurring during expiration [164] and the Porges–Bohrer method [165], which involves removing all sources of variance within a time series other than that which is attributable to breathing using a polynomial filter before computing the natural log of the variance of short time intervals of this filtered data.

RSA was measured in five studies. Both Patron, et al. [166] and Munafò, et al. [167] observed significant improvements in cardiac surgery patients only after the intervention, and not in comparators. In contrast, Schumann, et al. [145], Hjelland, et al. [168] and Lewis, et al. [133] observed no effect of the HRV-BF group or comparators with two of these studies [133,145] using healthy populations.

Overall, findings with regard to the effect of HRV-BF on RSA are mixed, possibly due to the small number of studies. On average, those studies finding an effect achieved a higher quality assessment score (Good) relative to those finding no effect, or an alternative effect direction (Fair), which may indicate that the direction of effect is influenced by study quality. Nevertheless, this conclusion is caveated by the small number of studies using this outcome. A summary of the methodological characteristics of all studies using this outcome measure can be found in Table 11.

#### 3.2.10. Conclusions—Physiological Measures

Overall, across indices, the effects of HR(V)-BF appear to be mixed though there are stronger trends towards no effect of the intervention for HF-HRV, RSA, and pNN50 and trends towards an effect for coherence measures.

### 3.3. Neural Measures

Four studies incorporated a neural measure. A summary of the methodological characteristics of all studies using neural outcome measures can be found in Table 12.

Three [128,140,169] utilised healthy populations, whilst Caldwell [95] studied major depressive disorder.

Schumann, et al. [140] found significant increases in resting functional connectivity between the ventromedial prefrontal cortex, and the middle cingulate cortex and posterior and anterior insular, as well as increases in positive functional connectivity in network nodes within the central autonomic network including the anterior cingulate cortex only in the HRV-BF group post-intervention. Moreover, changes in prefrontal connectivity to the left anterior insular correlated with increases in SDNN. Comparably, Kotozaki, et al. [169] observed increased anterior cingulate cortex and prefrontal cortex regional gray matter variation, after biofeedback of HR and cerebral blood flow but not in no-intervention controls. In their analysis, Nashiro, et al. [128] investigated changes in functional network connectivity of the canonical network categories identified by Laird, et al. [170] following HRV-BF or sham HRV-BF. These authors found significant increases in functional connectivity of the emotion/interoception network category after HRV-BF relative to all other categories (motor/visuospatial, visual, and cognitive). Moreover, in exploratory analyses of each canonical network, significant increases in functional connectivity were observed in Network 1 (consisting of the primary olfactory and limbic association cortices) and Network 5 (consisting of the midbrain) following HRV-BF relative to the comparator. Both of these networks are implicated in interoceptive functions and are contained within the emotion/interoception network category. In addition, one other canonical category (Network 16) also showed greater increases in functional connectivity after HRV-BF than sham HRV-BF. This network contains the posterior insular cortex, a region which is implicated in interoception [171]. Therefore, these authors concluded that HRV-BF increases the functional connectivity of brain networks implicated in the processing of interoceptive information.

In contrast, Caldwell [95] observed no change in ACC resting state connectivity at 2 weeks post-intervention for any group despite significant increases in SDNN in the HRV-BF group.

#### Conclusions—Neural Measures

These findings suggest that HR(V)-BF increases central autonomic network functional connectivity; however, effects do not appear to extend to clinical populations. Interestingly, the average quality assessment score for those studies finding an effect was higher (Good) than those observing no effect of the intervention (Fair) with a large absolute point difference. Nevertheless, the small number of studies utilising neural outcome measures may be skewing this finding, and as such, this observation should be treated with caution.

### 3.4. Overall Results

Overall summary statements for each of the outcome measures can be found in Table 13. brainsci-14-00579-t013_Table 13Table 13Overall summary of the results for each outcome measure.OutcomeIncluded Studies (*n*)Overall Summary StatementBehavioural1No effect of HR(V)-BFPhysiological

HF-HRV46Trend towards no effect of HR(V)-BFLF-HRV41Split results but trend for effects to be more frequently observed in healthy populations.RMSSD42Trend towards no effect of HR(V)-BFSDNN51Mixed results appear to be affected by covariance and compliance.pNN5012Trend towards no effect of HR(V)-BF.TP14Mixed findingsCoherence5Mixed findings but trend towards an effect of HR(V)-BF.BRS5Mixed results.RSA5Split results, but trend favouring more towards no effect.Neural4Trend towards an effect of HR(V)-BF amongst healthy populations.

## 4. Discussion and Conclusions

The purpose of this review was to ascertain the current state of knowledge with respect to the effects of HR(V)-BF therapy on behavioural, physiological, and neural outcome measures linked to interoception. Overall, a paucity of research studies utilised behavioural or neural measures, limiting conclusions for these measurement modalities. In contrast, a significant proportion of included studies utilised physiological measures related to interoception, particularly HF-HRV, LF-HRV, SDNN, and RMSSD. Below, we discuss these findings and provide recommendations for future investigation based on these observations. In addition, we propose a potential mechanism underlying how HR(V)-BF may affect interoception.

### 4.1. Behavioural Measures

For behavioural measures, the one included study found no effect of HR(V)-BF. However, using the heartbeat counting task [22], a measure confounded by time estimation strategies (i.e., counting seconds) and guessing based on knowledge of one’s own heartbeat [172] amongst other methodological weaknesses (see [173] for a review), may have impacted results. Corroborating this assumption, improvements in the heartbeat discrimination task (an alternative behavioural interoceptive measure) following a resonance frequency breathing protocol have been observed [174]. Therefore, it is possible that the lack of effect of HR(V)-BF observed here may be attributable to the interoceptive task used.

Strikingly, no studies have incorporated measures of interoceptive awareness or sensibility [1]. Given that there is increasing recognition of the significance of interoceptive awareness and sensibility for emotion regulation [175] and mental health [176,177], the lack of studies considering these as outcome measures represents a significant gap in the current knowledge. Consequently, conclusions regarding the effects of HR(V)-BF on behavioural measures of interoception are currently limited by an overall lack of empirical research and a reliance upon less reliable measures of this construct.

### 4.2. Physiological Measures

In this review, enhancements in indices of parasympathetic activity (e.g., higher HF-HRV, RMSSD, or pNN50), or overall autonomic activity (e.g., higher SDNN or TP) at rest post-intervention were regarded as indirect measures of interoception insofar as these indices reflect improved autonomic regulation which, presumably, depends upon the accurate perception and integration of interoceptive afferents within the brain. With respect to these measures, results were mixed. Specifically, for most measures, the data leaned towards a lack of effect of HR(V)-BF, particularly for HF-HRV, RMSSD, pNN50, and RSA. For coherence measures, there was a trend towards an effect, whereas for SDNN, BRS, TP, and LF-HRV, there was considerable heterogeneity in the results observed. Notably, HF-HRV positively correlates with interoceptive sensibility [40] and is typically reduced in clinical populations with interoceptive deficits (see [178]). Hence, given that most studies did not observe improvements in HF-HRV, or other indices of parasympathetic activity, this could suggest HR(V)-BF is not an effective interoceptive intervention.

Nevertheless, whilst during regular breathing HF-HRV is mainly composed of RSA [179], when engaged in resonance frequency breathing, the respiration rate is lowered, causing a shift in RSA to the LF-HRV band [61]. Therefore, during resonance frequency breathing, parasympathetic activity is better indexed by LF-HRV. Yet, mixed findings were observed for LF-HRV, meaning no clear conclusion can be drawn as to whether autonomic regulation is enhanced following HR(V)-BF therapy. Moreover, included studies only measured HRV at rest when participants were not expected to engage in resonance frequency breathing; hence, LF-HRV may not index parasympathetic activity in this context. Nevertheless, it is possible some participants still intentionally slowed their breathing at post-intervention measurements which may account for some of the variability in LF-HRV results and is worthy of further exploration.

Critically, no clear improvements in RMSSD or pNN50, time-domain measures of parasympathetic activity that are less influenced by respiration rate [179,180], were observed. Thus, taken together, these findings could suggest that HR(V)-BF is not effective in improving interoception. Yet, some included studies reported the presence of moderators which may be important for understanding the heterogeneity in observed findings. First, exploratory analyses by both Schumann, et al. [145] and Tinello, et al. [148] found that improvements in RMSSD (and SDNN for Tinello, et al. [148]) were only present in participants with low RMSSD at baseline. This finding implies the presence of a ceiling effect where participants with higher baseline parasympathetic activity are less likely to benefit from HR(V)-BF therapy. Consequently, future studies should consider stratifying participants according to their baseline levels of autonomic activity to avoid effects being masked by those with already high autonomic control.

Compliance with the intervention may also be an important moderator. Limmer, et al. [91] controlled for compliance by excluding participants who did not complete the minimal number of prescribed sessions, whereas Ratajczak, et al. [121] included a per-protocol sample containing only those participants who achieved resonance across sessions. Critically, Limmer, et al. [91] found improvements in SDNN and Ratajczak, et al. [121] observed improvements in SDNN and LF-HRV, as well as RMSSD only in the compliant groups. Therefore, compliance with the resonance frequency protocol may be necessary to observe effects in some indices related to interoception following HR(V)-BF.

Moreover, both Limmer, et al. [91] and Ratajczak, et al. [121] utilised intense treatment protocols. Therefore, it is possible that resonance is required to be achieved over many sessions before an effect on physiological indices is observed. Supporting this conclusion, for indices of both general autonomic activity such as SDNN and TP, and centrally mediated indices such as HF-HRV, LF-HRV, and RMSSD, there is a trend towards those studies finding an effect to be of higher intensity but also more likely to use a resonance frequency breathing protocol (though this trend was more apparent in some variables more than others). Most notably, for HF-HRV, whilst most studies did not find an intervention effect, those that did were more intense (100% of studies finding an effect were intense interventions vs. 33.33% of those who found an effect) and a greater proportion used some form of resonance frequency breathing (87.50% studies finding an effect used resonance breathing vs. 57.58% of those findings no effect). Similarly, the number of studies using an intense treatment protocol was much higher for those finding an effect for RMSSD, (61.54% vs. 17.86%). However, as this review is only qualitative, conclusions are not supported by statistical evidence. Therefore, caution should be applied when interpreting these findings and further studies should aim to empirically validate these assumptions.

Overall, though a large proportion of included studies utilised physiological measures related to interoception, heterogeneity in the results limits conclusions as to the effects of HR(V)-BF on these outcomes. Nevertheless, there is evidence to suggest that when resonance is achieved consistently over a number of sessions, effects are more likely to be observed. We discuss this observation further below.

### 4.3. Neural Measures

When considering neural measures, there was evidence to suggest that HR(V)-BF improves the functional connectivity of interoceptive brain regions contained within the central autonomic network. Specifically, changes in the functional connectivity of the anterior and posterior insula [128,140] as well as the cingulate cortex [140,169] were observed. Both the insula and anterior cingulate cortex have important roles in the integration of interoceptive afferents and the coordination of appropriate efferent responses (see [13] for a discussion). Hence, changes in the connectivity of these regions could imply that HR(V)-BF may enhance the brain’s ability to sense and regulate interoceptive signals. Nevertheless, the fact that these measures were not taken whilst the individual performed an interoceptive task, nor were they correlated with performance on a behavioural interoceptive task, could mean that changes in functional connectivity may relate to the engagement of these regions in other functions aside from interoception. For example, the anterior cingulate and insula are implicated in interoception [15], but also emotional regulation [181] and autonomic regulation [28]. Yet, models such as the neurovisceral integration model [182] consider interoception, autonomic regulation, and emotional regulation as inherently interlinked through their recruitment of the central autonomic network. Consequently, even if these changes in functional connectivity reflect improved autonomic, or emotional regulation, interoceptive improvements may potentially occur alongside these. Nonetheless, without corroboration from behavioural interoceptive measures, conclusions regarding the effects of HR(V)-BF on neural interoceptive measures are tentative.

Interestingly, Schumann, et al. [140] observed a correlation between SDNN and changes in functional connectivity between the prefrontal cortex and the anterior insula. As this is a correlational analysis, it cannot be determined whether (a) HR(V)-BF enhanced the intensity of afferent communication (possibly through stimulation of the vagus nerve) which led to subsequent elevations in overall autonomic activity and changes in functional connectivity as a result of this increased stimulation, or (b) whether HR(V)-BF stimulated improvements in functional connectivity which in turn facilitated enhanced regulation of interoceptive afferents and thus observed increases in SDNN reflect reflect enhanced parasympathetic activity. However, given that neither Schumann, et al. [140] nor Nashiro, et al. [128] observed improvements in RMSSD, an indicator of parasympathetic activity, following HR(V)-BF the latter possibility is unlikely. Consequently, whilst simultaneous increases in functional connectivity of interoceptive brain regions and afferent signalling intensity have been observed following HR(V)-BF, without improvements in autonomic regulation, it is not clear that the integration and evaluation of incoming interoceptive afferents have been improved by the intervention. Nevertheless, both Schumann, et al. [140] and Nashiro, et al. [128] used a moderate intervention structure, and therefore it is possible that, over longer durations, other changes may be evident, a concept discussed further below.

Whilst those studies that observed improvements in functional connectivity were with non-clinical populations, Caldwell [95] observed improvements in both LF-HRV and SDNN following HR(V)-BF, but not in functional connectivity of interoceptive brain regions when using a sample with major depression. Major depression is associated with pathologically decreased connectivity [183] and abnormalities of resting state activity [184] within the insula and cingulate cortices. Hence, within clinical populations, HR(V)-BF may increase the strength of afferent signalling, but changes in functional connectivity may be more difficult to achieve as a result of neuropathologies. Indeed, for centrally mediated indices (i.e., RMSSD, pNN50, HF-HRV, and LF-HRV), there was a trend for studies utilising clinical populations to be less likely to observe an effect, whereas for indices of overall autonomic activity (i.e., TP, or SDNN), clinical populations were more evenly distributed across those finding an effect and those who did not.

Consequently, there is some evidence to suggest HR(V)-BF may produce improvements in the functional connectivity of interoceptive brain regions. However, whether this translates into improvements in interoception (indexed by enhanced autonomic regulation, or through correlation with behavioural measures) is not yet clear. Moreover, it is possible that HR(V)-BF therapy may be less effective in enhancing indices related to interoception in clinical populations, a finding which has significant implications for its therapeutic potential.

### 4.4. A Proposed Mechanism

Overall, across behavioural, physiological, and neural outcome measures of interoception, there is mixed evidence to support an effect of HR(V)-BF, with a large proportion of findings leaning towards a lack of effect. However, we note that there was large heterogeneity amongst studies with respect to the population used, treatment intensity, and breathing protocol employed which limits the conclusions that can be drawn. Despite this, we identified two potential trends in the evidence base (1) effects on interoceptive outcomes are more likely to be observed when the protocol is more intense, and resonance frequency breathing is adhered to, and (2) effects on functional connectivity and indices of parasympathetic activity are less apparent in clinical populations. Based on these findings, we propose a speculative mechanism by which HR(V)-BF may plausibly improve interoception. In proposing this model, we emphasise that, due to the heterogeneity in both the methods and results of studies included within this review, and the proportion of evidence in support of this proposition, this model is speculative and further research is required to support, or refute, its claims.

This review was motivated by our hypothesis that HR(V)-BF may serve to improve interoception via its proposed stimulation of the vagus nerve [65]. Interestingly, similar to the HR(V)-BF literature, there is also large heterogeneity with respect to the effect of subcutaneous and transcutaneous vagus nerve stimulation on vagally mediated indices of HRV, with many studies observing no effect on measures such as RMSSD (see [185] or [186] for a review), whereas there is more evidence to support an effect of vagus nerve stimulation on indices of overall afferent signalling intensity (e.g., SDNN) (e.g., [187]) (see [186] for a review). Whilst, for HR(V)-BF, results were more mixed with respect to effects on SDNN, we found that effects were more apparent when variables which could influence the intensity with which the vagus nerve is stimulated during HR(V)-BF (namely, overall compliance, and compliance with resonance frequency breathing [91,121]) were controlled. With traditional vagus nerve stimulation, the device administers a set stimulation intensity and hence, providing individuals attend sessions, there is less likelihood that the amount of vagus nerve stimulation will vary across participants. Therefore, this may explain why results for SDNN are less heterogenous amongst vagus nerve stimulation interventions.

Moreover, as with HR(V)-BF, several studies have shown that vagus nerve stimulation increases the functional connectivity of interoceptive brain regions [68,69]. Critically, vagus nerve stimulation also has been shown to enhance performance on behavioural measures of interoception, even when no improvements in RMSSD are observed [71]. Thus, this could suggest that changes in behavioural and neural indices of interoception may be able to occur without changes in vagal physiological indices. Therefore, given this finding, and the overall similarities between the current findings and those of the vagus nerve stimulation literature, it is possible that HR(V)-BF may improve interoception, though further research investigating whether changes in SDNN and neural indices following HR(V)-BF are corroborated by behavioural or neural indices of interoceptive awareness (i.e., heartbeat-evoked potentials) is required. Based on these observations, we propose a three-stage model of how HR(V)-BF may improve interoception.

In our model, each stage is thought to occur sequentially over time. In the first stage, resonance frequency breathing induces oscillatory coherence between bodily systems [63] which, in turn, is thought to increase the intensity of afferent signalling via stimulation of the vagus nerve [65]. Resultantly, at this first stage, we would expect to see a general amplification in indices of autonomic activity such as SDNN, or TP, especially in those studies where resonance frequency breathing is used. This trend was observed in this review, with greater effects on SDNN and TP being evident in those studies which implemented resonance breathing, particularly those that controlled for compliance [121].

At the second stage, we propose that this amplification of vagus nerve activity, and subsequent stimulation of the autonomic nervous system enhances the ability of the central autonomic network to detect physiological afferents, thus resulting in increased communication and connectivity between brain regions implicated in autonomic regulation. Indeed, when considering neural measures, both Schumann, et al. [140] and Nashiro, et al. [128] found changes in the connectivity of interoceptive networks, as well as improvements in SDNN. Whilst the directionality of this relationship cannot be determined, it seems logical that an increase in afferent signalling intensity would increase the activity of the central autonomic network and therefore impact the organisation of brain regions implicated in the detection of these signals. In turn, at this stage, due to the increased detection of interoceptive signals, we would also expect to see improvements in behavioural measures of interoception.

In the final stage, this strengthening and reorganisation of connections within cortical interoceptive structures is proposed to have long-term effects on the regulation of autonomic activity. Specifically, it is at this point we would expect to see improvements in indices of parasympathetic activity, including HF-HRV, RMSSD, and pNN50. Indeed, for all these indices, but particularly for HF-HRV, we noted a trend for those studies which found an effect (albeit a small number) to have higher proportions of resonance frequency breathing protocols and to be of greater treatment intensity. We account for this observation by incorporating aspects of predictive coding [188] and interoceptive inference [189] within our model.

Specifically, within predictive coding models, the brain generates hypotheses about the causes of incoming sensory input from the environment [190]. These predictions are then used to coordinate actions which serve to regulate internal and external processes and maintain equilibrium [188]. Hence, the ultimate goal of a living system is to form an accurate self-model from which reliable predictions and action behaviours can be generated [191]. This, in turn, requires the minimisation of free energy [192], or ‘prediction errors’ which occur when there is a misalignment between one’s predictions, and the sensory input that has been received. When prediction errors occur, these are mitigated either by changing the nature of the sensory information such that it aligns with one’s predictions (e.g., through engaging reflex arcs), or through updating one’s predictions such that these will be more accurate when the same context is encountered again [189].

Importantly, whether predictions are updated is dependent upon two characteristics of the sensory signal, (a) its precision, or how much weight is given to that sensory signal (see [193] for a discussion), and (b) the accumulation of evidence, or how many repeated instances of this prediction error have occurred [194]. In situations where a signal is highly salient, and thus given high precision, or where there is an accumulation of prediction errors over time, then prior updating will occur [195]. Our three-stage model aligns with this prior updating process: first, there is the increased salience of afferent signalling which is detected by the brain and registered as a prediction error; then, this highly salient signal is experienced over multiple treatment sessions, triggering the accumulation of evidence necessary for updating predictions; finally, the updating of predictions leads to changes in the regulation of autonomic activity by the central autonomic network. Consequently, framing our model through the lens of predictive coding can help to provide a plausible explanation as to why interventions using resonance frequency protocols and higher intensities were more likely to find an effect. In essence, these intervention structures provide both the sensory salience and the accumulation of sensory evidence necessary for the updating of priors and the restoration of optimal autonomic functioning.

In turn, our model can also provide a framework for understanding why HR(V)-BF appears to be less effective in improving autonomic regulation in clinical populations. Specifically, in certain clinical conditions, afferent interoceptive signals are postulated to be ‘noisy’ [2], leading to the generation of a chronic prediction error. In response to this, the weighting given to interoceptive streams can be maladaptively altered, resulting in the development of pathologies. For example, it has been proposed that with conditions such as eating disorders [9], noisy interoceptive signals result in these signals becoming attenuated, rendering individuals increasingly reliant upon predictions. However, due to interoceptive attenuation, these predictions become increasingly inaccurate and fail to respond adaptively to changes in the autonomic state [196]. The net result is a state of reduced autonomic flexibility characterised by lower heartrate and aberrantly high HRV.

Consequently, improving interoception in clinical conditions requires reducing afferent noise. Through resonance frequency breathing, HR(V)-BF may improve the predictability of afferent signals by constraining and amplifying oscillations [197] at a frequency of ~0.1 Hz, thus increasing signal-to-noise ratios of afferent signals being communicated via the vagus nerve [174] (see also [71]). However, to override aberrant precision-weighting mechanisms in clinical conditions, more intense, cross-context learning experiences may be required than healthy individuals without afferent noise. Therefore, this would explain why clinical conditions were less likely to observe improvements in neural indices and centrally mediated physiological indices following HR(V)-BF as it may be that they require more repeated instances with resonance in order to successfully update predictions and restore an optimal self-model. Therefore, our model suggests that HR(V)-BF can confer benefits for clinical populations when the treatment intensity is sufficient and resonance frequency breathing is used. Nevertheless, given that clinical populations are more likely to struggle with adherence to interventions [198,199], if the efficacy of HR(V)-BF depends upon strict compliance with an intense treatment regimen, then this may significantly limit its potential as an intervention for clinical conditions with interoceptive dysfunctions.

In sum, our three-stage model makes three key hypotheses:Over the course of HR(V)-BF therapy, improvements will first occur in indices of central autonomic activity, such as SDNN and TP, followed by changes in neural connectivity and behavioural measures and then by indices of autonomic regulation.Higher compliance with resonance breathing will be associated with an increased likelihood of intervention effects at each of these stages.The greater the number of total intervention sessions, and the greater intensity with which they are delivered, the more likely there is for changes to be observed across all interoceptive indices.

### 4.5. Directions for Future Research

First, we note that only a small proportion of included studies utilised intense intervention structures (i.e., more than one session per week) in combination with a resonance frequency breathing protocol. Therefore, in order to better understand the moderating effects of these variables on outcomes related to interoception (and to test the second and third hypotheses from our model), future research should consider how the effects of HR(V)-BF on the interoceptive indices considered in this review vary according to the intensity of the intervention and the breathing protocol used.

More specifically, investigations where the degree to which a participant is breathing at their resonance frequency during HR(V)-BF may serve to elucidate the extent to which resonance frequency breathing is necessary for bringing about interoceptive improvements following HRV-BF.

In addition, as compliance also appears to serve as a potential moderator of the effects of HR(V)-BF on interoception, we encourage future investigations to better report compliance with the intervention overall, particularly with respect to home practice. In this review, many included studies incorporated home practice using HR(V)-BF apps and technologies. However, as compliance with home practice was rarely reported, we did not take into account the amount of home practice when categorising a study’s intervention intensity (unless only home practice was used) as it was not certain whether clients adhered to home practice instructions. Furthermore, by reporting compliance to home interventions one can better ascertain whether self-guided HR(V)-BF is an effective intervention for interoception and thus whether it may help to ease the increasing burden on therapists within healthcare systems worldwide.

Furthermore, many studies did not incorporate a long-term follow-up measure. Consequently, understanding with regard to the long-term impact of HR(V)-BF on interoception is limited. Therefore, we echo prior conclusions that future studies should incorporate assessments of the long-term carry-over effects of HR(V)-BF (e.g., Wheat and Larkin, 2010), particularly with clinical populations where the aim is to achieve enduring symptom improvement.

Interoception is implicated in the wider concept of body awareness which refers to one’s conscious representation of the self formed from the integration of interoceptive and exteroceptive information and modified by cognitive processes including appraisals, emotions, memories, and beliefs [200]. In some neurological disorders, changes to both body awareness and interoception have been observed (e.g., in multiple sclerosis; [201]). Therefore, if HR(V)-BF is found to improve interoception, then incorporating HR(V)-BF into neurorehabilitation protocols may aid in bringing about improvements in symptoms and quality of life for people living with neurological conditions (see [202] for a discussion).

Moreover, recent studies have shown that interoception is positively associated with time perception [203]. In turn, it has been proposed that measuring individuals’ perceived duration of interoceptive stimuli could potentially provide insight into the level of noise within an interoceptive stream (see [204]). Future studies drawing upon the methodology of Di Lernia, et al. [204] could therefore provide a more sensitive measure of how predictive processes change following HR(V)-BF and hence whether HF(V)-BF can restore optimal precision of interoceptive signalling, as proposed by our model.

### 4.6. Limitations of Included Studies

We note some key limitations of the data included within this review:Overall, the average quality score of included studies was ‘Fair’ and this may have contributed to the observed heterogeneity in findings. However, we note that we expected (and found) the majority of studies to employ a clinical trial structure and therefore chose a bias assessment tool designed to evaluate these studies. Consequently, this may have affected the quality scores of studies not employing this structure.Some studies did not adhere to The European Society of Cardiology and The North American Society of Pacing and Electrophysiology Task Force [49] standards of HRV measurement and reporting and therefore heterogeneity in findings may arise from inconsistencies in the degree to which potential confounds were controlled for across studies. For example, though medications such as antipsychotic and antidepressant medications do not impact HRV, others such as tricyclic antidepressants and clozapine reduce HRV below normal levels (see [205] for a review), whereas medications such as beta-blockers can induce positive changes in HRV [206]. Therefore, medications may confound treatment effects. Future studies should ensure all medications are clearly reported and, where possible, controlled for in analyses.Only one study incorporated HR-BF [169], rather than HRV-BF. Furthermore, in this study, HR-BF was combined with the biofeedback of cerebral blood flow from the rostrolateral prefrontal cortex to the frontopolar cortex. Accordingly, there is currently limited evidence regarding the efficacy of HR-BF as an interoceptive intervention.As has been previously noted [207], we observed a lack of standardised intervention protocols, including the type of HR(V)-BF used (e.g., some studies used games, apps, or HR(V)-BF in combination with other interventions), and the comparator used which thus limits the conclusions that can be drawn. Hence, there is a pertinent need for studies to adhere to a standardised procedure for conducting HR(V)-BF.Variability in whether a transformation was applied to HRV data, and the type of transform applied, was observed between studies. Therefore, it may be important for future studies to investigate whether the type of transformation applied changes the result observed.Many studies did not report the treatment protocol in sufficient detail to facilitate replication, particularly with respect to the breathing techniques used (e.g., no reporting of the ratio of inhalation to exhalation used). Consequently, we echo the call by Lalanza, et al. [84] for more standardised and transparent reporting in investigations of HR(V)-BF.

### 4.7. Limitations of the Review Process

There are also some limitations to note with the review process:As this review assessed the current state of the literature on the effects of HR(V)-BF on interoception, we used broad inclusion criteria. Consequently, there was heterogeneity amongst included studies, meaning a quantitative synthesis was not conducted and therefore conclusions do not have statistical support. Moreover, as most studies did not report effect sizes, our conclusions are based on *p*-values which may be inflated by sample sizes. Therefore, we encourage future reviews to employ more stringent inclusion criteria in order to facilitate the conducting of meta-analyses which could provide further evidence for, or against, our conclusions.Joint screening of only 10 percent of papers at the level of full-text may also be viewed as a limitation. Yet, inter-rater agreement at both title-and-abstract and full-text screening was substantial; therefore, it is unlikely that papers will have been missed using this approach. Relatedly, as only one author conducted data extraction, this may have increased the propensity for errors. However, the data extraction table was thoroughly checked by both an independent reviewer, and the reviewer responsible for data extraction. Therefore, errors in this table are unlikely.Conference abstracts included within this review could not be assessed for risk of bias and hence the quality of the evidence obtained in these investigations is unclear. Nevertheless, including these studies allowed us to provide a comprehensive overview of the literature, including the grey literature sources.

### 4.8. Conclusions

Overall, in the current knowledge state, there is mixed evidence supporting HR(V)-BF as an intervention for interoception when considering findings across behavioural, physiological, and neural outcomes related to interoception.

#### 4.8.1. Behavioural Measures

For behavioural measures, only one included study utilized this outcome; therefore, without further research incorporating measures across behavioural interoceptive dimensions, no clear conclusion can be drawn and hence we call for more research to be conducted in this area.

#### 4.8.2. Physiological Measures

Concerning physiological measures, mixed findings were observed with a trend towards indices of parasympathetic activity (e.g., HF-HRV or RMSSD) being less likely to observe an effect, particularly amongst clinical populations. Nevertheless, compliance with the intervention (particularly the resonance frequency protocol) and the intensity of the intervention were both identified as potential moderators of the effects of HR(V)-BF on physiological measures related to interoception. Therefore, findings suggest that with sufficient treatment intensity and compliance with a resonance frequency breathing protocol, HR(V)-BF may be an effective intervention for the improvement of these outcomes.

#### 4.8.3. Neural Measures

Considering neural measures, HR(V)-BF was found to improve the functional connectivity of brain regions implicated in interoception, at least in healthy populations. Therefore, these findings suggest HR(V)-BF has promise as an intervention for the improvement of neural interoceptive indices. However, future research correlating neural improvements with behavioural interoceptive measures is required to corroborate this conclusion.

#### 4.8.4. Concluding Remarks

We have proposed a model which outlines how HR(V)-BF could improve interoception which can account for the pattern of findings observed in this review. Whilst currently the potential of HR(V)-BF as an interoceptive intervention remains unclear, we encourage researchers to follow our recommendations and directions for future research to provide further support for, or against, the potential of HR(V)-BF as an intervention for interoceptive deficits.

## Figures and Tables

**Figure 1 brainsci-14-00579-f001:**
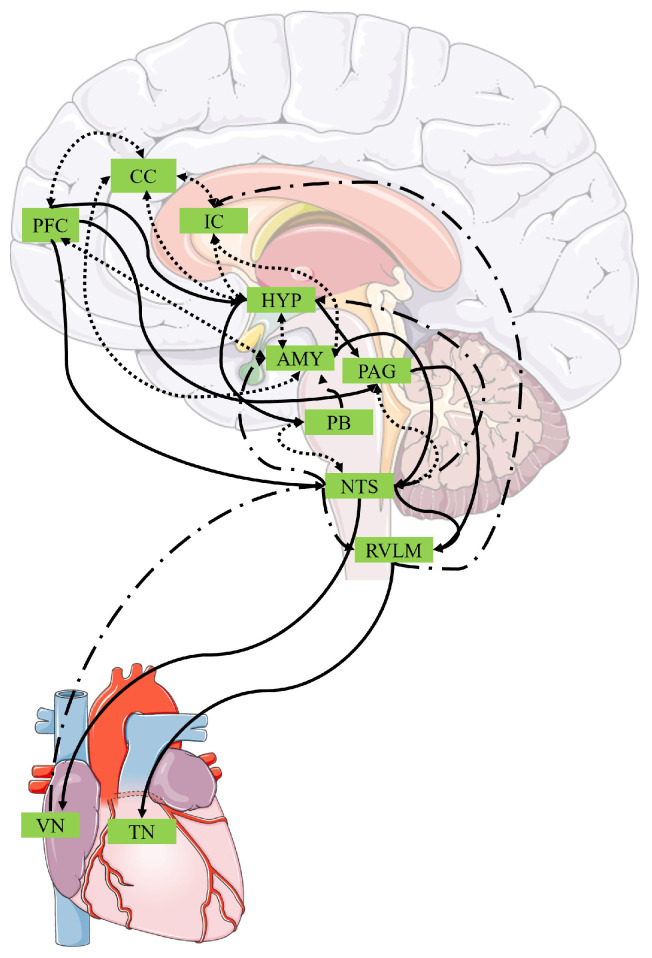
Diagram depicting the different afferent and efferent pathways involved in the detection and regulation of cardiovascular activity by the central autonomic network. Solid black arrows depict efferent pathways from brain to heart. Dot-and-dashed lines indicate afferent pathways from the heart to the brain. Dotted black arrows depict bi-directional communication. AMY = amygdala, CC = cingulate cortex, HYP = hypothalamus, IC = insular cortex, NTS = nucleus of the solitary tract, PAG = periaqueductal grey, PB = parabrachial nucleus, PFC = prefrontal cortex, RVLM = rostral ventrolateral medullary neurons, TN = thoracic nerves, VN = vagus nerve. Diagram adapted from Ellis and Thayer [32]. This diagram has been simplified for ease of understanding. For all brain regions implicated in cardiovascular interoception, please refer to the diagram by Ellis and Thayer [32]. This figure was partly generated using images from Servier Medical Art, provided by Servier, licensed under a Creative Commons Attribution 3.0 Unported License.

**Figure 2 brainsci-14-00579-f002:**
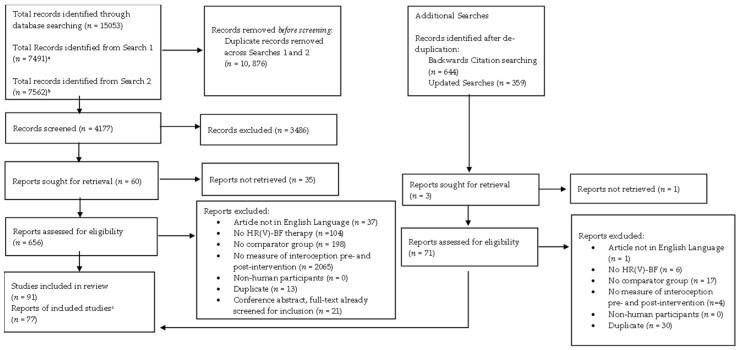
Flowchart depicting the screening process. Flowchart adapted from [75]. ^a^ Academic Search Ultimate (*n* = 1072), PsycINFO (*n* = 1481), MEDLine (*n* = 1106), CINAHL (*n* = 618), SCOPUS (*n* = 2085), Web of Science (*n* = 1129). ^b^ Academic Search Ultimate (*n* = 1084), PsycINFO (*n* = 1487), MEDLine (*n* = 1121), CINAHL (*n* = 625), SCOPUS (*n* = 2101), Web of Science (*n* = 1144). ^c^ Some studies meeting criteria were not reported on the basis that the conference abstract contained insufficient data to be included and the full-text was not published (*n* = 13).

**Table 1 brainsci-14-00579-t001:** Definitions and included measures of each of the interoceptive measurement modalities (behavioural, physiological, or neural).

Interoceptive Measurement Modality	Definition	Included Measures
Behavioural	Any measure or task assessing individuals’ conscious experience of interoceptive signals including measures of interoceptive accuracy, awareness, or sensibility.	Heartbeat counting task, heartbeat discrimination task, heartbeat detection task, validated questionnaire measures of interoceptive awareness, or sensibility (e.g., the Multidimensional Assessment of Interoceptive Awareness; [79]), measures of interoceptive sensibility or awareness (e.g., confidence ratings of performance on behavioural tasks).
Physiological	Measures of cardiac visceral afferent signalling which reflect autonomic regulation by the central autonomic network, or improvements in afferent communication to the brain.	HRV frequency and time-domain indices, indices of baroreflex functioning (e.g., baroreflex sensitivity), indices of resonance or coherence (e.g., coherence index), measures of RSA.
Neural	Measures of the neural representation of interoceptive information. This includes the representation of physiological afferents in the brain, patterns of brain activation during attention to interoceptive signals, functional connectivity of interoceptive brain regions, or the relation of brain structure to interoception at other processing levels.	Functional connectivity of the central autonomic network or interoceptive brain regions, activation of interoceptive brain regions (e.g., the insular cortex), electrical activity representing the interoceptive signal (e.g., the heartbeat-evoked potential).

**Table 2 brainsci-14-00579-t002:** Summary of the key features of included studies utilising behavioural measures, including methodological characteristics (whether a clinical population was employed, type of comparator group, intensity of intervention, and type of breathing protocol) as well as mean risk of bias scores, grouped according to the effect observed. Methodological characteristics of studies finding each effect are reported as a proportion of all studies finding the same effect.

Features of Included Studies	Effect of Intervention (*n* = 0)	No Effect of Intervention (*n* = 1)	Other Effect (i.e., Time or Effect of Comparator)
Clinical population (%)	-	0	-
Comparator Group (%)	-	NI—0 TAU—0 AC—100 S/P—0	-
Intervention Intensity (%)	-	Mild—100 Moderate—0 Intense—0	-
Resonance Frequency Breathing (%)	-	Overall RF—100.00 Optimal—100.00 Individual—0.00 Paced/preset—0.00 Unclear—0.00	-
Risk of Bias (Mean)	-	19	-

Note. NI = No intervention, TAU = intervention-as-usual, AC = active control, S/P = sham/placebo, Overall RF = total proportion of resonance frequency protocols (individual or optimal).

**Table 3 brainsci-14-00579-t003:** Summary of the key features of included studies utilising HF-HRV, including methodological characteristics (whether a clinical population was employed, type of comparator group, intensity of intervention, and type of breathing protocol) as well as mean risk of bias scores, grouped according to the effect observed. Methodological characteristics of studies finding each effect are reported as a proportion of all studies finding the same effect.

Features of Included Studies	Effect of Intervention (*n* = 8)	No Effect of Intervention (*n* = 33)	Other Effect (*n* = 5) (i.e., Time or Effect of Comparator)
Clinical population (%)	50.00	69.70	40.00
Comparator Group (%) ^a^	NI—30.00 TAU—20.00 AC—30.00 S/P—20.00	NI—41.67 TAU—25.00 AC—25.00 S/P—8.33	NI—20.00 TAU—40.00 AC—20.00 S/P—20.00
Intervention Intensity (%)	Mild—0.00 Moderate—0.00 Intense—100.00	Mild—30.30 Moderate—36.36 Intense—33.33	Mild—40.00 Moderate—40.00 Intense—20.00
Resonance Frequency Breathing (%)	Overall RF—87.50 Optimal—50.00 Individual—37.50 Paced/preset—12.50 Unclear—0.00	Overall RF—57.58 Optimal—39.40 Individual—18.19 Paced/preset—24.24 Unclear—18.19	Overall RF—80.00 Optimal—60.00 Individual—12.5 Paced/preset—12.50 Unclear—0.00
Risk of Bias (Mean)	14.50	14.52	15.00

Note. NI = No intervention, TAU = intervention-as-usual, AC = active control, S/P = sham/placebo, Overall RF = total proportion of resonance frequency protocols (individual or optimal). ^a^ Some studies used more than one comparator group; therefore, proportion has been calculated out of the total number of comparators for the main effect.

**Table 4 brainsci-14-00579-t004:** Summary of the key features of included studies utilising LF-HRV, including methodological characteristics (whether a clinical population was employed, type of comparator group, intensity of intervention, and type of breathing protocol) as well as mean risk of bias scores, grouped according to the effect observed. Methodological characteristics of studies finding each effect are reported as a proportion of all studies finding the same effect.

Features of Included Studies	Effect of Intervention (*n* = 19)	No Effect of Intervention (*n* = 18)	Other Effect (*n* = 4) (i.e., Time or Effect of Comparator)
Clinical population (%)	47.37	72.22	75.00
Comparator Group (%) ^a^	NI—36.36 TAU—22.73 AC—22.73 S/P—18.18	NI—47.37 TAU—21.05 AC—21.05 S/P—10.53	NI—60.00 TAU—20.00 AC—20.00 S/P—0.00
Intervention Intensity (%)	Mild—31.58 Moderate—36.84 Intense—31.58	Mild—33.33 Moderate—22.22 Intense—44.44	Mild—0.00 Moderate—50.00 Intense—50.00
Resonance Frequency Breathing (%)	Overall RF–63.16 Optimal—52.63 Individual—10.53 Paced/preset—31.58 Unclear—5.26	Overall RF—50.00 Optimal—38.39 Individual—16.17 Paced/preset—22.22 Unclear—22.22	Overall RF—100.00 Optimal—25.00 Individual—75.00 Paced/preset—0.00 Unclear—0.00
Risk of Bias (Mean)	15.05	14.28	13.50

Note. NI = No intervention, TAU = intervention-as-usual, AC = active control, S/P = sham/placebo, Overall RF = total proportion of resonance frequency protocols (individual or optimal). ^a^ Some studies used more than one comparator group; therefore, proportion has been calculated out of the total number of comparators for the main effect.

**Table 6 brainsci-14-00579-t006:** Summary of the key features of included studies utilising SDNN, including methodological characteristics (whether a clinical population was employed, type of comparator group, intensity of intervention, and type of breathing protocol) as well as mean risk of bias scores, grouped according to the effect observed. Methodological characteristics of studies finding each effect are reported as a proportion of all studies finding the same effect.

Features of Included Studies	Effect of Intervention (*n* = 26)	No Effect of Intervention (*n* = 21)	Other Effect (*n* = 4) (i.e., Time or Effect of Comparator)
Clinical population (%)	65.38	61.90	0.00
Comparator Group (%) ^a^	NI—34.48 TAU—24.14 AC—34.48 S/P—6.90	NI—39.13 TAU—21.74 AC—26.09 S/P—13.04	NI—80.00 TAU—0.00 AC—20.00 S/P—0.00
Intervention Intensity (%)	Mild—26.92 Moderate—34.62 Intense—38.46	Mild—38.10 Moderate—33.33 Intense—28.57	Mild—50.00 Moderate—25.00 Intense—25.00
Resonance Frequency Breathing (%)	Overall RF—73.08 Optimal—50.00 Individual—23.08 Paced/preset—19.23 Unclear—7.69	Overall RF—66.67 Optimal—42.86 Individual—23.80 Paced/preset—23.81 Unclear—9.52	Overall RF—50.00 Optimal—25.00 Individual—25.00 Paced/preset—50.00 Unclear—0.00
Risk of Bias (*M*)	14.85	15.05	15.00

Note. NI = No intervention, TAU = intervention-as-usual, AC = active control, S/P = sham/placebo, Overall RF = total proportion of resonance frequency protocols (individual or optimal). ^a^ Some studies used more than one comparator group; therefore, proportion has been calculated out of the total number of comparators for the main effect.

**Table 7 brainsci-14-00579-t007:** Summary of the key features of included studies utilising pNN50, including methodological characteristics (whether a clinical population was employed, type of comparator group, intensity of intervention, and type of breathing protocol) as well as mean risk of bias scores, grouped according to the effect observed. Methodological characteristics of studies finding each effect are reported as a proportion of all studies finding the same effect.

Features of Included Studies	Effect of Intervention (*n* = 3)	No Effect of Intervention (*n* = 8)	Other Effect (*n* = 1) (i.e., Time or Effect of Comparator)
Clinical population (%)	33.33	62.50	0.00
Comparator Group (%) ^a^	NI—33.33 TAU—33.33 AC—33.33 S/P—0.00	NI—37.50 TAU—12.50 AC—37.50 S/P—12.50	NI—0.00 TAU—0.00 AC—100.00 S/P—0.00
Intervention Intensity (%)	Mild—66.67 Moderate—0.00 Intense—33.33	Mild—25.00 Moderate—37.50 Intense—37.50	Mild—100.00 Moderate—0.00 Intense—0.00
Resonance Frequency Breathing (%)	Overall RF—66.67 Optimal—33.33 Individual—33.33 Paced/preset—0.00 Unclear—33.33	Overall RF—87.50 Optimal—50.00 Individual—37.50 Paced/preset—12.50 Unclear—0.00	Overall RF—0.00 Optimal—0.00 Individual—0.00 Paced/preset—0.00 Unclear—0.00
Risk of Bias (Mean)	14.67	14.63	16.00

Note. NI = No intervention, TAU = intervention-as-usual, AC = active control, S/P = sham/placebo, Overall RF = total proportion of resonance frequency protocols (individual or optimal). ^a^ Some studies used more than one comparator group; therefore, proportion has been calculated out of the total number of comparators for the main effect.

**Table 8 brainsci-14-00579-t008:** Summary of the key features of included studies utilising TP, including methodological characteristics (whether a clinical population was employed, type of comparator group, intensity of intervention, and type of breathing protocol) as well as mean risk of bias scores, grouped according to the effect observed. Methodological characteristics of studies finding each effect are reported as a proportion of all studies finding the same effect.

Features of Included Studies	Effect of Intervention (*n* = 8)	No Effect of Intervention (*n* = 7)	Other Effect (*n* = 0) (i.e., Time or Effect of Comparator)
Clinical population (%)	75.00	57.14	-
Comparator Group (%) ^a^	NI—20.00 TAU—30.00 AC—20.00 S/P—30.00	NI—37.50 TAU—25.00 AC—25.00 S/P—12.50	-
Intervention Intensity (%)	Mild—12.50 Moderate—25.00 Intense—62.50	Mild—14.29 Moderate—42.86 Intense—42.86	-
Resonance Frequency Breathing (%)	Overall RF—50.00 Optimal—37.50 Individual—12.50 Paced/preset—37.50 Unclear—12.50	Overall RF—42.86 Optimal—28.57 Individual—14.29 Paced/preset—42.86 Unclear—14.29	-
Risk of Bias (Mean)	15.38	13.86	-

Note. NI = No intervention, TAU = intervention-as-usual, AC = active control, S/P = sham/placebo, Overall RF = total proportion of resonance frequency protocols (individual or optimal). ^a^ Some studies used more than one comparator group; therefore, proportion has been calculated out of the total number of comparators for the main effect.

**Table 9 brainsci-14-00579-t009:** Summary of the key features of included studies utilising coherence ratios, including methodological characteristics (whether a clinical population was employed, type of comparator group, intensity of intervention, and type of breathing protocol) as well as mean risk of bias scores, grouped according to the effect observed. Methodological characteristics of studies finding each effect are reported as a proportion of all studies finding the same effect.

Features of Included Studies	Effect of Intervention (*n* = 3)	No Effect of Intervention (*n* = 2)	Other Effect (*n* = 0) (i.e., Time or Effect of Comparator)
Clinical population (%)	66.67	50.00	-
Comparator Group (%) ^a^	NI—75.00 TAU—0.00 AC—25.00 S/P—0.00	NI—0.00 TAU—50.00 AC—50.00 S/P—0.00	-
Intervention Intensity (%)	Mild—33.33 Moderate—33.33 Intense—33.33	Mild—100.00 Moderate—0.00 Intense—0.00	-
Resonance Frequency Breathing (%)	Overall RF—66.67 Optimal—0.00 Individual—66.67 Paced/preset—33.33 Unclear—0.00	Overall RF—0.00 Optimal—0.00 Individual—0.00 Paced/preset—100.00 Unclear—0.00	-
Risk of Bias	14.33	16.00	-

Note. NI = No intervention, TAU = intervention-as-usual, AC = active control, S/P = sham/placebo, Overall RF = total proportion of resonance frequency protocols (individual or optimal). ^a^ Some studies used more than one comparator group; therefore, proportion has been calculated out of the total number of comparators for the main effect.

**Table 10 brainsci-14-00579-t010:** Summary of the key features of included studies utilising BRS, including methodological characteristics (whether a clinical population was employed, type of comparator group, intensity of intervention, and type of breathing protocol) as well as mean risk of bias scores, grouped according to the effect observed. Methodological characteristics of studies finding each effect are reported as a proportion of all studies finding the same effect.

Features of Included Studies	Effect of Intervention (*n* = 3)	No Effect of Intervention (*n* = 2)	Other Effect (*n* = 0) (i.e., Time or Effect of Comparator)
Clinical population (%)	33.33	50.00	-
Comparator Group (%) ^a^	NI—50.00 TAU—0.00 AC—50.00 S/P—0.00	NI—0.00 TAU—0.00 AC—550.00 S/P—0.00	-
Intervention Intensity (%)	Mild—0.00 Moderate—75.00 Intense—25.00	Mild—0.00 Moderate—50.00 Intense—50.00	-
Resonance Frequency Breathing (%)	Overall RF—100.00 Optimal—100.00 Individual—0.00 Paced/preset—0.00 Unclear—0.00	Overall RF—50.00 Optimal—50.00 Individual—0.00 Paced/preset—50.00 Unclear—0.00	-
Risk of Bias	12.33	14.00 ^b^	-

Note. NI = No intervention, TAU = intervention-as-usual, AC = active control, S/P = sham/placebo, Overall RF = total proportion of resonance frequency protocols (individual or optimal). ^a^ Some studies used more than one comparator group; therefore, proportion has been calculated out of the total number of comparators for the main effect. ^b^ Lehrer and Vaschillo [160] were not included in this average due to being a conference abstract and therefore have no risk of bias score.

**Table 11 brainsci-14-00579-t011:** Summary of the key features of included studies utilising RSA, including methodological characteristics (whether a clinical population was employed, type of comparator group, intensity of intervention, and type of breathing protocol) as well as mean risk of bias scores, grouped according to the effect observed. Methodological characteristics of studies finding each effect are reported as a proportion of all studies finding the same effect.

Features of Included Studies	Effect of Intervention (*n* = 2)	No Effect of Intervention (*n* = 3)	Other Effect (*n* = 0) (i.e., Time or Effect of Comparator)
Clinical population (%)	50.00	33.33	-
Comparator Group (%) ^a^	NI—0.00 TAU—50.00 AC—50.00 S/P—0.00	NI—66.67 TAU—0.00 AC—33.33 S/P—0.00	- - - -
Intervention Intensity (%)	Mild—0.00 Moderate—50.00 Intense—50.00	Mild—66.67 Moderate—33.33 Intense—0.00	- - -
Resonance Frequency Breathing (%)	Overall RF—100.00 Optimal—0.00 Individual—100.00 Paced/preset—0.00 Unclear—0.00	Overall RF—33.33 Optimal—33.33 Individual—0.00 Paced/preset—66.67 Unclear—0.00	- - - - -
Risk of Bias	15.50	14.33	-

Note. NI = No intervention, TAU = intervention-as-usual, AC = active control, S/P = sham/placebo, Overall RF = total proportion of resonance frequency protocols (individual or optimal). ^a^ Some studies used more than one comparator group; therefore, proportion has been calculated out of the total number of comparators for the main effect.

**Table 12 brainsci-14-00579-t012:** Summary of the key features of included studies utilising neural measures, including methodological characteristics (whether a clinical population was employed, type of comparator group, intensity of intervention, and type of breathing protocol) as well as mean risk of bias scores, grouped according to the effect observed. Methodological characteristics of studies finding each effect are reported as a proportion of all studies finding the same effect.

Features of Included Studies	Effect of Intervention (*n* = 3)	No Effect of Intervention (*n* = 1)	Other Effect (*n* = 0) (i.e., Time or Effect of Comparator)
Clinical population (%)	0.00	100.00	-
Comparator Group (%) ^a^	NI—33.33 TAU—0.00 AC—33.33 S/P—33.33	NI—0.00 TAU—100.00 AC—0.00 S/P—0.00	-
Intervention Intensity (%)	Mild—33.33 Moderate—33.33 Intense—33.33	Mild—0.00 Moderate—100.00 Intense—0.00	-
Resonance Frequency Breathing (%)	Overall RF—66.67 Optimal—66.67 Individual—0.00 Paced/preset—0.00 Unclear—33.33	Overall RF—100.00 Optimal—100.00 Individual—0.00 Paced/preset—0.00 Unclear—0.00	-
Risk of Bias	16.33	13.00	-

Note. NI = No intervention, TAU = intervention-as-usual, AC = active control, S/P = sham/placebo, Overall RF = total proportion of resonance frequency protocols (individual or optimal). ^a^ Some studies used more than one comparator group; therefore, proportion has been calculated out of the total number of comparators for the main effect.

## Data Availability

Supporting data for this review, including the anonymised protocol, search strategy, screening criteria, screening decision logs, quality assessment, coding of methodological characteristics, and data extraction table are openly available on the Open Science framework within the Appendix A.

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
