# Peer review of "The Utility of Heartrate and Heartrate Variability Biofeedback for the Improvement of Interoception across Behavioural, Physiological and Neural Outcome Measures: A Systematic Review"

_brainsci, 2024, doi:10.3390/brainsci14060579_

Round 1
Reviewer 1 Report
Comments and Suggestions for Authors
This review reporting the association of interoceptive measures with physiological variables is interesting and wellwritten. It will be useful to researchers starting their activity in the field of the relation between interoception and health.
There is something the authors may with to consider
1 major pointt
since there is a difference in results obtained using absolute or normalized spectral measures of HRV, it could be useful to clarify which of these measures were employed in the considered papers. This could be one of the factors contributing to mixed results
Minor points
In the version of the manuscript I downloaded from the website, the quality of the figures is poor. Can the authours do something to improve it?
Reviewer 2 Report
Comments and Suggestions for Authors
Thank you for this paper. The paper is well-written and is clear in almost all cases. The quality of the methodology of this systematic review is very high, and the authors described the its process in detail. The paper is not short, therefore, some edits should be made to increase the readability.
1. Due to the fact that this paper has a lot of aspects, it would be beneficial to add subsections into the introduction.
2. It would beneficial to present some subsections in the discussion in the form of a table (e.g., limitations). Overall, future directions seem to be presented twice (subsections 4.1 and 4.2). It would be useful to synthesize the information. "Entities are not to be multiplied beyond necessity” (William of Ockham).
3. Conclusions should be presented in a more comprehensive manner. Now, they seem somewhat poor and do not provide significant amount of information with regards to the potential of HR(V)-BF as an interoceptive in- tervention across behavioural, physiological and neural outcome measures related to interoception.
4. I feel that some subsections are overlapped (i.e., subsections 2.3 and 2.4). Please make sure if there is a need to split up this material. Overall, please look at the paper as a whole, and make sure that there are no overlapped subsections or duplicated materials.
5. Lines 407-411 (Table 2) and other tables. Please provide more details about what data you present here and how you assessed these. Especially, the first column "Methodological characteristic" is unclear. Please describe what you mean here.
6. Please make sure that all footnotes for tables are presented. In some cases, footnote with "b" were not presented (e.g., Table 3).
7. Subsections 3.1.1 and 3.2.10 have the same title "Conclusions". Please make sure that these titles are specific. Please reconsider, for example, by editing this into "Conclusions on smth...".
Thank you for your paper. In my opinion, it represents a really high level of methodology.
Round 2
Reviewer 2 Report
Comments and Suggestions for Authors
Thank you for your improvements. I have only one comment, in general it is a previous comment:
Conclusions should be presented in a more comprehensive manner. Now, they seem somewhat poor and do not provide significant amount of information with regards to the potential of HR(V)-BF as an interoceptive intervention across behavioural, physiological and neural outcome measures related to interoception.
Now, I have stressed in the sentence (see above) what I especially mean here. Please present separately specific conclusions on behavioural, physiological and neural outcome measures (separately for these 3 categories).
